

# Water-use characteristics of *Syzygium antisepticum* and *Adinandra integerrima* in a secondary forest of Khao Yai National Park in Thailand with implications for environmental management

Ratchanon Ampornpitak[1,2], Anuttara Nathalang[3] and Pantana Tor-ngern[4,5]

[1] International Program in Hazardous Substance and Environmental Management, Graduate School, Chulalongkorn University, Bangkok, Thailand
[2] Center of Excellence on Hazardous Substance Management, Chulalongkorn University, Bangkok, Thailand
[3] National Biobank of Thailand, National Science and Technology Development Agency, Pathum Thani, Thailand
[4] Department of Environmental Science, Faculty of Science, Chulalongkorn University, Bangkok, Thailand
[5] Water Science and Technology for Sustainable Environment Research Unit, Chulalongkorn University, Bangkok, Thailand

Corresponding author
Pantana Tor-ngern,
pantana.t@chula.ac.th

## ABSTRACT

**Background.** Southeast Asia has experienced widespread deforestation and change in land use. Consequently, many reforestation projects have been initiated in this region. However, it is imperative to carefully choose the tree species for planting, especially in light of the increasing climate variability and the potential alteration of plantation on the watershed water balance. Thus, the information regarding water-use characteristics of various tree species and sizes is critical in the tree species selection for reforestation.
**Methods.** We estimated tree water use ($T$) of dominant species including *Syzygium antisepticum* and *Adinandra integerrima,* hereafter *Sa* and *Ai*, respectively, in a secondary tropical forest in Khao Yai National Park, Thailand, using sap flow data, and compared $T$ between species and size classes. Additionally, we evaluated the responses of $T$ of both species in each size class to environmental factors including soil moisture and vapor pressure deficit (VPD).
**Results.** Results showed consistently higher $T$ in *Sa* compared to *Ai* across ranges of VPD and soil moisture. Under low soil moisture, $T$ of *Sa* responded to VPD, following a saturating exponential pattern while *Ai* maintained $T$ across different VPD levels, irrespective of tree size. No responses of $T$ to VPD were observed in either species when soil water was moderate. When soil moisture was high, $T$ of both species significantly increased and saturated at high VPD, albeit the responses were less sensitive in large trees. Our results imply that *Ai* may be suitable for reforestation in water-limited areas where droughts frequently occur to minimize reforestation impact on water availability to downstream ecosystems. In contrast, *Sa* should be planted in regions with abundant and reliable water resources. However, a mixed species plantation should be generally considered to increase forest resilience to increasing climate variation.

## INTRODUCTION

Over an annual timescale with negligible change in total water storage, precipitation is coarsely partitioned into evapotranspiration and runoff in the forest water cycle (*Wang & Tang, 2014*). Because tree water use ($T$) constitutes 40–90% of evapotranspiration (*Jasechko et al., 2013*; *Deb Burman et al., 2019*), the quantity of $T$ affects the amount of precipitation that ultimately contributes to runoff, impacting the downstream ecosystems. With the projected increases of global climate change impacts, $T$ may be altered through changes in environmental conditions including temperature and precipitation regimes, severity of weather and climate extremes such as droughts and floods (*Menezes-Silva et al., 2019*). Such changes will certainly affect the responses of $T$ to environmental factors, therefore governing the outflow from forests. Thus, evaluating such responses would provide insights into environmental management that involves water cycling, such as predicting runoff from forests which may result in floods or droughts in the downstream ecosystems.

The variations of $T$ are mainly related to tree size (*Meinzer et al., 2005*; *Jung et al., 2011*) and environmental factors including soil water availability, solar radiation, and vapor pressure deficit (VPD) which represents atmospheric humidity (*Xu & Yu, 2020*; *Gutierrez Lopez et al., 2021*). Several previous studies reported significant variations of $T$ with tree size. The relationship between tree diameter and $T$ was found among several species of angiosperms (*Meinzer et al., 2005*), *Eucalyptus crebra* and *Callitris glaucophylla* in evergreen woodland in Australia (*Zeppel & Eamus, 2008*), and trees in a temperate mixed-deciduous forest in South Korea (*Jung et al., 2011*). Additionally, different tree sizes have been linked to different responses to droughts with large trees being more vulnerable than small trees to drought because of greater exposure to atmospheric demand (*Bennett et al., 2015*; *Stovall, Shugart & Yang, 2019*). However, information of the effects of tree size on $T$ is still lacking in secondary tropical forests. The effects of environmental factors on $T$ vary in different forest types and regions. For example, when soil moisture is not limited, $T$ strongly responds to VPD, which increases when the air humidity decreases, and solar radiation in an old-growth spruce forest in the Ore Mountains, Germany (*Clausnitzer et al., 2011*). Under soil water stress, *Brum et al. (2018)* found that $T$ could decrease with increasing VPD during an extreme drought in an Amazonian tropical rainforest. On the other hand, *Spanner et al. (2022)* found that the sensitivity of $T$ to soil moisture varied with species, with some increasing and some decreasing during the dry period in an old-growth upland forest in the central Amazon. Thus, changing environmental conditions can alter the response patterns of $T$.

Forests in Southeast Asia provide a wide range of important ecosystem services to many people and communities. Unfortunately, these forests have been disrupted by widespread deforestation and land use change (*Stibig et al., 2014*; *Zeng et al., 2018*), resulting in various stages of forests in the same area (*Curtis et al., 2018*). In particular, the areas that were

previously used for agricultural purposes have been abandoned for several years, and naturally or artificially transformed into secondary forests. Consequently, many of the degraded forests may not contribute much to improving biodiversity and mitigating climate change through carbon dioxide removal from the atmosphere. Therefore, reforestation projects have emerged in many countries in the tropics, highlighting the use of native species to avoid competition with other native trees, which can help restore biodiversity and sequester carbon (*Hooper, Condit & Legendre, 2002*). However, planting more trees in existing secondary forests may raise some concerns because trees are potentially heavy water users and might deplete water resources (*Jackson et al., 2005*). Also, reforestation may not be desirable in certain areas because it may reduce water availability for the existing trees and increase the evapotranspiration rate (*Van Kanten & Vaast, 2006*) and thus leading to reductions in runoff (*Li, Xu & Sun, 2014*). With these regards, an appropriate selection of tree species for planting is among the priority tasks for forest restoration since species-specific water-use characteristics play an important role in changing the components of the forest hydrologic cycle (*Van Dijk & Keenan, 2007*). However, the availability of such information is still limited in tropical forests, especially in secondary ones. Hence, it is imperative to evaluate the response patterns of $T$ to environmental factors in secondary tropical forests that would offer necessary information on species-specific water-use characteristics.

Khao Yai National Park (KYNP) is a UNESCO world heritage site. Most of the areas of KYNP consist of a mosaic of different stages of vegetation succession with more than 60% of the forests undergoing different stages of regeneration while the remainder are old-growth forests. Thus, secondary forests in KYNP are important to biodiversity conservation and climate change mitigation through the regulation of atmospheric carbon. With these regards, this study was performed in a secondary tropical forest at KYNP, representing a young forest aged ~10 years. In this study site, the dominant tree species include *Syzygium antisepticum* and *Adinandra integerrima*. *Syzygium antisepticum* can be found as the dominant species in other tropical forests such as tropical evergreen swamp forests in Cambodia (*Theilade et al., 2011*), dry evergreen forests in northeastern Thailand (*Bunyavejchewin, 1999*) and tropical coastal sand dune in southern Thailand (*Marod et al., 2020*). *Adinandra integerrima* can be found in other parts of Thailand, such as Doi Inthanon National Park in the northern region (*Georgiadis, 2022*) and other countries in the tropics, such as Cambodia, China, Laos, and Vietnam (*Tagane et al., 2020*). Despite the widespread presence of these species in Thailand and neighboring countries of Southeast Asia, the information on the water-use characteristics of both species is still lacking. Therefore, this study aims to (1) estimate $T$ of *Syzygium antisepticum* and *Adinandra integerrima* in a secondary tropical forest in KYNP, and (2) evaluate the responses of $T$ to environmental factors of both species in different tree size classes. The collected data covered a period from 18 September 2020 to 26 November 2022, including a wide range of environmental conditions. The outcome of this study would improve the understanding of species-specific water-use characteristics in secondary forests which can support policy design on the management of tropical forests and water resources. In addition, findings

from this study may provide a recommendation for selecting appropriate tree species for forest restoration in the tropical region.

## MATERIALS & METHODS

### Study site and measurements of the environmental variables

The study was conducted in Khao Yai National Park, Thailand (14°26′31″N, 101°22′55″E). Khao Yai National Park covers an area of about 200 km$^2$ in Nakhon Ratchasima, Saraburi, Prachinburi and Nakhon Nayok Provinces in Thailand. This region is dominated by monsoon climate, where the dry season usually lasts from November to April and from May to October for the wet season (*Brockelman, Nathalang & Maxwell, 2017*). Based on recorded data between 1994–2018, the overall mean annual temperature was 22.4 °C. The mean annual rainfall was 2,100 mm. Khao Yai National Park is characterized by different stages of forest succession comprising primary forests and various stages of secondary forests. In this study, we performed the study in a secondary forest representing a young forest in Nakhon Nayok Province. The study site has an area of 2 ha and an age of approximately 10 years (*Chanthorn, Hartig & Brockelman, 2017*). Its mean canopy height is 15 m and its tree density of 1,226 trees ha$^{-1}$. The soil is gray-brown ultisol which was degraded by shifting agriculture by burning before regeneration (*Chanthorn et al., 2016*; *Chanthorn, Hartig & Brockelman, 2017*). The bulk density was 1.24 g cm$^{-3}$ and soil texture was sandy clay-loam with the sand contents of 64.4% and 56.4% as measured in September 2020 and February 2021, respectively (*Rodtassana et al., 2021*). In 2020, a 20 m tall tower was constructed for installing weather sensors above the forest canopy in the plot. Environmental conditions that influence $T$ including atmospheric humidity, solar radiation, and soil moisture have been continuously monitored since then. Air temperature ($T$, °C), relative humidity (RH, %), and photosynthetically active radiation (PAR, µmol m$^{-2}$ s$^{-1}$) were measured by a temperature and relative humidity probe (EE181-PT; Campbell Scientific, Logan, UT, USA) and a quantum sensor (LI190R-PT, Campbell Scientific), respectively. Soil moisture sensors (Water content reflectometer, CS616-PT-U; Campbell Scientific) were installed to monitor volumetric soil moisture at 5, 10, 15, and 30 cm depth because tree roots may access water from multiple depths in the soil (*Wang et al., 2019*). We randomized the points to install soil moisture sensors around the tower. Two soil moisture sensors were installed at each depth of 5, 10, and 15 cm. However, soil moisture at 30 cm depth was monitored by one soil moisture sensor because soil moisture in subsoil was less sensitive to changing environmental conditions than topsoil (*Rong et al., 2017*). Rainfall (mm) was measured by tipping rain gauge bucket (TE525MM-PT; Campbell Scientific). All sensors were connected to a datalogger (CR1000 series; Campbell Scientific, Logan, UT, USA) which recorded data every 30 min. Air temperature and relative humidity are used to calculate vapor pressure deficit (VPD, kPa), which is the difference between actual vapor pressure and saturated vapor pressure (SVP), from the following equations (*Monteith & Unsworth, 1990*).

$$SVP = 610.7 \times 10^{\frac{7.5T}{237.5+T}} \tag{1}$$

$$\text{VPD} = \left(1 - \frac{\text{RH}}{100}\right) \times \text{SVP}. \tag{2}$$

Because we did not have any information regarding rooting depth, which determines the depth of soil moisture data to be used in the analysis, we used the average of soil moisture data from all soil water probes, covering soil depth up to 30 cm, as the soil moisture data $(\theta, \text{m}^3\text{m}^{-3})$ for further analysis. Based on previous studies in the central Amazon which reported the most fine root distribution within 20 cm soil depth (*Noguchi et al., 2014*), we assumed that the average soil moisture across 30 cm depth represents the soil water that largely influences tree water use. To facilitate the cross-site comparison with other or future studies, relative extractable water (REW) was used in the analysis and was calculated according to *Granier, Loustau & Bréda (2000)*

$$\text{REW} = \frac{\theta - \theta m}{\theta \text{FC} - \theta m} \tag{3}$$

where $\theta$ is the average soil moisture of all sensors across 30-cm soil depth, $\theta_m$ is minimum volumetric soil moisture and $\theta_{FC}$ is the soil water at field capacity. In the plot where soil water at field capacity has not been measured, maximum volumetric soil moisture during the study period can be used as $\theta_{FC}$ for the REW calculation (*Tor-ngern et al., 2018*). Accordingly, we used the maximum and minimum $\theta_{average}$ that were determined from our data during the study period to represent $\theta_{FC}$ and $\theta_m$, respectively.

## Species selection and tree sampling

The tree species were chosen based on the relative abundance of basal area in this plot, which was calculated from the basal area of one species relative to total basal area of all species within the site. To examine the difference in tree water use, two dominant tree species with similar leaf phenology were selected for this study. As a result, *Syzygium antisepticum* and *Adinandra integerrima,* hereafter *Sa and Ai,* respectively, which have evergreen leaf habit, were chosen to measure water flow rate. We attempted to select trees to cover the range of size distribution within the plot, based on the inventory data from the site (W Chanthorn, pers. comm., 2018), by partitioning the tree size classes into 10-cm intervals and sampled three trees from each size class. However, due to the requirement of trees being within 25 m radius from the data logger, 4 trees of *Sa* and 5 trees of *Ai* were selected for the measurement (Table 1).

## Sap flux measurement and scaling up from the point measurement to whole-tree water use

Sap flux density ($J_s$), which represents water mass flowing through a unit area per time in trees, was measured using self-constructed thermal dissipation probes (TDPs) (*Granier, 1985*). Each TDP set contains one non-heated and one heated probe being supplied with a constant ~0.2 W electrical power. Before inserting TDPs into the stems, debarking around the drilling point was done before drilling the holes for TDP installation. Two holes were drilled with approximately 10–15 cm spacing between two probes. Based on previous studies in pine trees, the patterns of radial variation in $J_s$ along the sapwood depth were observed with higher $J_s$ in the outer sapwood layers than in the inner sapwood

**Table 1  Information of the selected study trees. DBH refers to the diameter at breast height in cm. Sapwood area (in cm$^2$) was estimated using an allometric equation derived from dominant species in the study site (*Yaemphum, Unawong & Tor-Ngern, 2022*).**

| Species | DBH (cm) | Sapwood area (cm$^2$) |
|---|---|---|
| *Adinandra integerrima* | 13.7 | 135.92 |
| *Adinandra integerrima* | 6.5 | 30.64 |
| *Adinandra integerrima* | 11 | 87.67 |
| *Adinandra integerrima* | 5.3 | 20.38 |
| *Adinandra integerrima* | 11.3 | 92.51 |
| *Syzygium antisepticum* | 24.8 | 444.88 |
| *Syzygium antisepticum* | 22.4 | 363.02 |
| *Syzygium antisepticum* | 18.7 | 253.09 |
| *Syzygium antisepticum* | 17.8 | 229.34 |

layers (*Ford et al., 2004*; *Oishi, Oren & Stoy, 2008*). Therefore, ignoring the radial variation of $J_s$ may produce an error when scaling up from $J_s$ to $T$. However, previous studies in tropical forests that use similar sap flow sensors only measured $J_s$ at the outer sapwood because of the unknown pattern of sapwood area in tropical tree species (*Horna et al., 2011*; *Raquel Salas-Acosta et al., 2022*). In addition, most tropical trees have diffuse-porous wood without distinct annual rings and tend to have a sap flow rate that is similar along the radial sapwood depth (*Lu, Urban & Ping, 2004*). Therefore, we assumed that $J_s$ was uniform along the sapwood depth of the selected trees when scaling from single-point measurements to the whole-tree level, and only measured $J_s$ at the outer 2-cm sapwood at breast height (~1.3 m above ground). In addition, azimuthal variation of $J_s$ may produce variation when scaling up from $J_s$ to $T$ (*Lu, Müller & Chacko, 2000*; *James et al., 2002*; *Tateishi et al., 2008*). This variation depends on the effect of forest canopy shading by neighboring trees. In other words, trees may be obstructed from sunlight by canopy shading from surrounding trees leading to varying $J_s$ along the circumference. In this study, the surrounding trees were equally distributed around the measured trees. Nevertheless, we installed two sensors in the north and the south directions in some trees which may be influenced by canopy shading at certain times during the day. Data from TDPs were recorded as 30-minute means of voltage difference between the probes ($\Delta V$, mV) by the same data logger (CR1000; Campbell Scientific, Logan, UT, USA) that recorded environmental data. For the analysis, the voltage difference was converted to $J_s$ (g m$^{-2}$ s$^{-1}$) using an empirical equation (*Granier, 1987*):

$$J_s = 118.99 \times 10^{-6} \times \left( \frac{\Delta V_m - \Delta V}{\Delta V} \right)^{1.231} \tag{4}$$

where $\Delta V_m$ is the maximum voltage difference under no flow conditions which usually occurs at night and when VPD is low. The Baseliner program version 4.0 was used to select $\Delta V_m$ to calculate $J_s$ (*Oishi, Hawthorne & Oren, 2016*). The program automatically determines the maximum daily $\Delta V$ to represent $\Delta V_m$. Maximum voltage difference may occur every night if air humidity is very high, or VPD reaches 0 kPa, resulting in potentially

zero water flow. However, this assumption is not valid for many ecosystems due to nighttime transpiration (*e.g.*, *Caird, Richards & Donovan, 2007*; *Forster, 2014*; *Dayer et al., 2020*) or recharge of stem water (*Phillips & Oren, 1998*). For these reasons, no universal rule exists for identifying $\Delta V_m$. The Baseliner software takes an approach to $\Delta V_m$ estimation by first identifying points in time where flow is likely zero and allowing the user to visually inspect and modify those points.

To scale up from $J_s$ to $T$, we employed the following approach. Daily sum $J_s$ (g m$^{-2}$ day$^{-1}$) was considered in the analysis to avoid issues related to the nighttime recharge of stem water that may increase as soil moisture becomes more depleted (*Phillips & Oren, 1998*). When nighttime recharge increases with decreasing soil moisture, the proportions of sap flux at night relative to sap flux during the day become larger. This problem can be avoided when calculating $T$ as a daily sum (*Phillips & Oren, 1998*). For trees with sensors in the north and the south direction, daily sum $J_s$ from both sensors were averaged to a mean daily $J_s$ ($J_{mean}$) for each of them (*Kunert et al., 2012*). The following equation was used to estimate $T$:

$$T = 1800 \times 10^{-7} \times J_{mean} \times A_S \tag{5}$$

where $T$ is daily tree water use (L d$^{-1}$), $J_{mean}$ is mean daily sum $J_s$ (g m$^{-2}$ day$^{-1}$) and $A_S$ is sapwood area (cm$^2$). In both species, $A_S$ was estimated based on an allometric equation which was derived from 13 dominant species in an old growth and a secondary forest (the same plot as this study site) at Khao Yai National Park as follows (*Yaemphum, Unawong & Tor-Ngern, 2022*):

$$y = 0.728 \times^{1.998} \tag{6}$$

where $y$ is sapwood area (cm$^2$), $x$ is diameter at breast height (cm).

### Data analysis

For the analysis, we used the environmental data and $T$ between 18 September 2020 to 26 November 2022. The data covered two years which represents a wide range of environmental conditions. To avoid the potential effects of wet canopy conditions that may inhibit $T$ when the leaf surface is covered with water droplets (*Aparecido et al., 2016*), we selected the days under rain-free conditions to perform the analysis.

To evaluate the responses of $T$ to environmental factors including VPD and REW, we performed a boundary line analysis (*Schäfer, Oren & Tenhunen, 2000*) to obtain the response of $T$ to environmental factors under non-limiting conditions. Trees were categorized based on the size distribution of each species as presented in Table 1 into large trees (DBH $\geq$ 10 cm for *Ai* and DBH $\geq$ 20 cm for *Sa*) and small trees (DBH <10 cm for *Ai* and DBH <20 cm for *Sa*). This results in 2 trees for both species in the small class, and three *Ai* trees and two *Sa* trees in the large class. After that, $T$ from all trees in the same category was averaged to mean $T$ ($T_{mean}$) for each day. Tree water use varies with VPD, REW and PAR (*Phillips & Oren, 2001*). Based on our data during the study period, VPD and PAR were highly correlated ($r = 0.79$, $p \leq 0.001$), therefore we focused on VPD and REW as environmental driving variables. We performed boundary line analysis after partitioning

data into three REW classes based on the REW distribution including low soil moisture (REW < 0.1), intermediate soil moisture (REW 0.1−0.4), and high soil moisture (REW > 0.4). With two classes of tree size (large and small), we had six subsets of data in both species. Each subset was subjected to the boundary line, designed to select data representing the maximum $T_{mean}$ for each tree size in each REW class along the range of VPD. The upper boundary line was derived by (1) partitioning $T_{mean}$ data of each REW class into at least five VPD intervals for appropriate number of data points in regression analysis (at least five data points per analysis), (2) calculating the mean and standard deviation of $T_{mean}$ in each interval, (3) removing outliers using Dixon's test, (4) selecting the data falling above the mean plus one standard deviation and (5) averaging the selected data for each VPD interval. For each tree size and REW class, the mean $T_{mean}$ values of all VPD intervals obtained in step (5) were analyzed by regression analysis. All regression analyses were performed in SigmaPlot version 12.0 (Systat Software, Inc., San Jose, CA USA). Data management and analysis were performed with Rstudio, version 1.3.1073 (*RStudio Team, 2020*).

# RESULTS

## Environmental conditions in the study site

During the study period, there were 52% rainy and 48% rain-free days. The average daily VPD and PAR inversely corresponded with rainfall, being low when rainfall occurred and vice versa. The maximum and minimum values of PAR during the study period were 575 and 57.3 $\mu$mol m$^{-2}$ s$^{-1}$, respectively, with an average of 345.76 $\pm$103.47 $\mu$mol m$^{-2}$ s$^{-1}$. The average daily VPD was 0.34 $\pm$0.23 kPa. Volumetric soil moisture of all depths was averaged into $\theta_{average}$. The maximum and minimum $\theta_{average}$ during the study period were 0.2 and 0.04 m$^3$m$^{-3}$, respectively. The $\theta_{average}$ was then used to calculate REW with an average value of 0.44 $\pm$ 0.25. Figure 1 summarizes the environmental conditions during the study period.

## Tree water use of *Syzygium antisepticum* and *Adinandra integerrima*

Tree water uses of both species during the study period are shown in Fig. 2. The average *T* values with one standard deviation of *Sa* and *Ai* were 21.48 $\pm$ 7.73 and 10.01 $\pm$ 4.04 L d$^{-1}$, respectively. Comparing *T* between both species, we found that the *T* of *Sa* was significantly higher than that of *Ai* under high soil moisture and high light conditions ($p < 0.0001$).

## Responses of tree water use to environmental factors in different tree size classes

Figure 3 summarizes the results of the responses of *T* to VPD under various REW ranges, with the regression statistics in Table 2. At low soil moisture (REW < 0.1, black circles), *T* of *Sa* increased with increasing VPD and gradually saturated at high VPD while that of *Ai* did not respond to the changing VPD, regardless of tree size. Under intermediate soil moisture conditions (REW 0.1− 0.4, gray squares), the *T* of both species in both sizes did not respond to VPD. Under high soil moisture (REW > 0.4, red triangles), the *T* of both

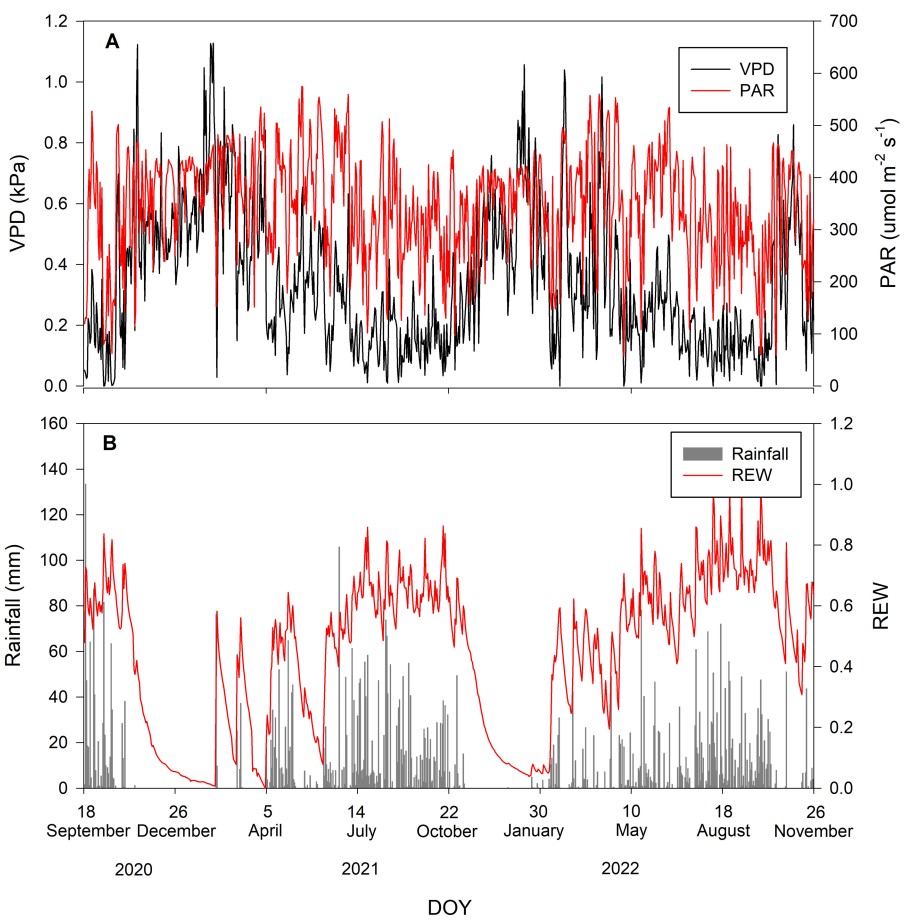

**Figure 1** **Environmental conditions during the study period.** Daily values of (A) vapor pressure deficit (VPD in kPa; black line) and photosynthetically active radiation (PAR in $\mu$mol m$^{-2}$ s$^{-1}$; red line) and (B) rainfall (mm; gray bar) and relative extractable water (REW; red line).

species in both sizes followed the saturating exponential pattern as previously described in the case of low soil moisture. However, the sensitivity of increasing $T$ at low VPD was different between the species and size class. In both species, $T$ of large trees was less sensitive to rising VPD than small ones (Table 2).

## DISCUSSION

Overall, the environmental data during the study period represent a wide range of environmental conditions which facilitates the analysis of $T$ responses to the environments. The maximum $T$ of $Sa$ in our data (47.54 L d$^{-1}$) was higher than the values that were found in $T$ of *Syzygium cordatum* in a peat swamp forest in South Africa (*Clulow et al., 2013*), ranging from 30 L d$^{-1}$ in the winter to 45 L d$^{-1}$ in the summer. Moreover, our average $T$ of Sa was within the range of $T$ found in *Eugenia natalitia* (2 to 28 L d$^{-1}$), which is the same family as $Sa$, as reported by the same study. Although we did not find studies that reported $T$ values of $Ai$ or similar genus, $T$ of $Ai$ was within the range of $T$ (10 to 1,180 L d$^{-1}$)
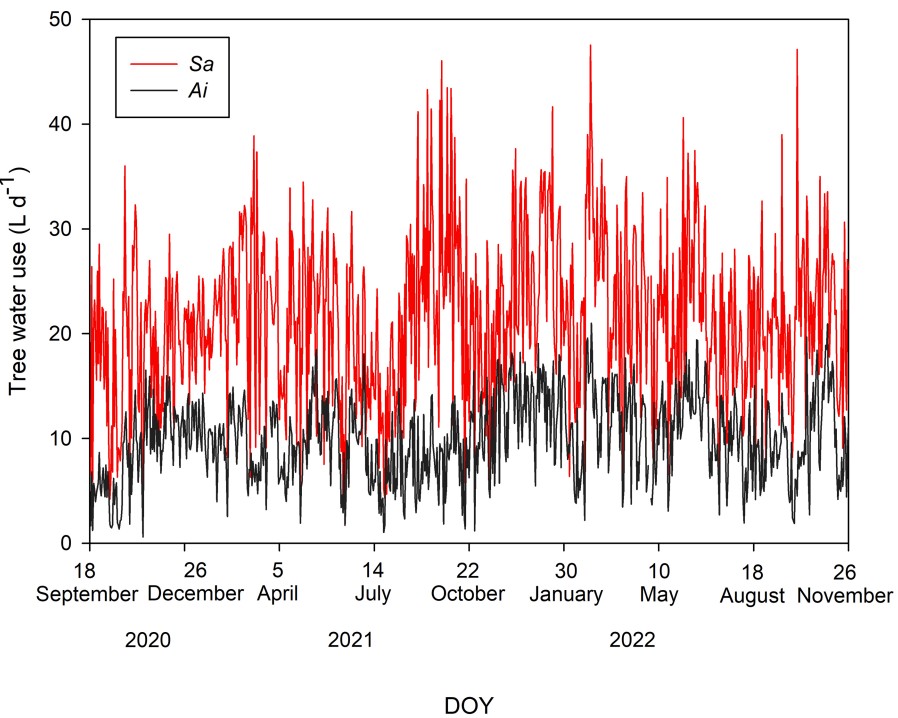

**Figure 2** **Daily tree water use.** Daily sum of tree water use (L d$^{-1}$) of *Syzygium antisepticum* (red line) and *Adinandra integerrima* (black line), averaged across all trees for each species, throughout the study period from 18 September 2020 to 26 November 2022.

**Table 2** **Summary of regression statistics.**

| Species | Size classes | REW classes | Relationships | $r^2$ | $p$ |
|---|---|---|---|---|---|
| *Syzygium antisepticum (Sa)* | Small (DBH <20 cm) | <0.1 | $T = 22.04 \times (1 - e^{-5.03 \times VPD})$ | 0.77 | 0.049 |
| | | 0.1–0.4 | n/a | 0.14 | 0.538 |
| | | >0.4 | $T = 27.82 \times (1 - e^{-8.55 \times VPD})$ | 0.80 | 0.042 |
| | Large (DBH ≥ 20 cm) | <0.1 | $T = 56.63 \times (1 - e^{-4.30 \times VPD})$ | 0.76 | 0.023 |
| | | 0.1–0.4 | n/a | 0.44 | 0.56 |
| | | >0.4 | $T = 51.34 \times (1 - e^{-6.75 \times VPD})$ | 0.85 | 0.027 |
| *Adinandra integerrima (Ai)* | Small (DBH <10 cm) | <0.1 | n/a | 0.13 | 0.868 |
| | | 0.1–0.4 | n/a | 0.21 | 0.440 |
| | | >0.4 | $T = 6.65 \times (1 - e^{-7.53 \times VPD})$ | 0.96 | 0.003 |
| | Large (DBH ≥ 10 cm) | <0.1 | n/a | 0.45 | 0.143 |
| | | 0.1–0.4 | n/a | 0.11 | 0.589 |
| | | >0.4 | $T = 29.25 \times (1 - e^{-5.08 \times VPD})$ | 0.99 | <0.0001 |

**Notes.**
$T$ is tree water use (Ld−1), VPD is vapor pressure deficit (kPa) and $r^2$ is the coefficient of determination and $p$ value for each regression result. The analyses were based on significance level of 0.05. n/a indicates no significant relationship was found.

found in 93 tree species from 52 reviewed publications that estimated whole-plant water use for trees growing in worldwide natural forests or plantations (*Wullschleger, Meinzer & Vertessy, 1998*). The study reported that the rates of water use ranged from 10 L d$^{-1}$ for

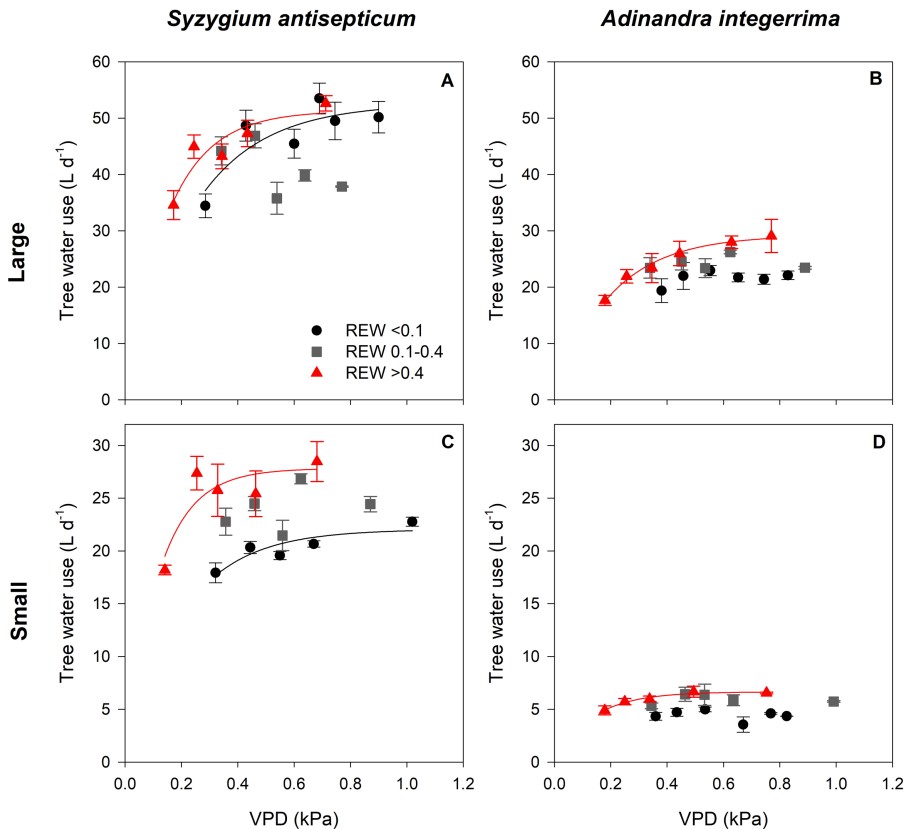

**Figure 3** **Relationship between tree water use and vapor pressure deficit under different soil moisture conditions.** Relationship between tree water use ($T$; L d$^{-1}$) and vapor pressure deficit (VPD; kPa) of *Syzygium antisepticum* in (A) large and (C) small size classes and *Adinandra integerrima* in (B) large and (D) small size classes under low soil moisture condition (REW ¡0.1, circles), under intermediate soil moisture condition (REW 0.1–0.4, squares), and under high soil moisture condition (REW > 0.4, triangles).

trees in a 32-year-old plantation of *Quercus petraea* L. ex Liebl. in eastern France to 1,180 L d$^{-1}$ for an overstory tree, *Euperua purpurea* Bth., growing in the Amazonian rainforest. Overall, the $T$ values of both species in this study were within the wide ranges found in previous studies in tropical settings (Table 3).

Previous studies showed that the variation of J$_s$ among trees of different ages and sizes is relatively low (*Kumagai et al., 2007*; *Jaskierniak et al., 2016*); thus, sapwood area may be a major determinant of $T$ in this study. Based on our data, J$_s$ of trees was similar between both species ($p = 0.278$), suggesting the greater contribution of sapwood area or tree size to the significant difference in $T$. Additionally, higher water use in large trees may imply their deeper access to groundwater whereas small trees may only consume water from shallow soil as previously shown in a study investigating water use by *Acer saccharum Marsh*. in different sizes (*Dawson, 1996*). Moreover, other research in tropical forests reported that large trees consume much more water relative to small trees as indicated by the positive relationship between water consumption and tree size (*O'Grady, Eamus & Hutley, 1999*; *Meinzer, Goldstein & Andrade, 2001*; *Horna et al., 2011*; *Aparecido et al., 2016*).

Ampornpitak et al. (2023), *PeerJ*, DOI 10.7717/peerj.16525

**Table 3  Summary of water use of tropical forest trees reviewed in this study.**

| Reference | Location | | Species | Method | DBH (cm) | Tree height (m) | T (Ld−1) | Period |
|---|---|---|---|---|---|---|---|---|
| Dye (1996) | Frankfort State Forest in South Africa (24°49′S, 30°43′E) | Site 1 (3-year-old trees) | Eucalyptus grandis | HPM | 14.7 ± 1.17 | 14.7 ± 1.01 | 45.8 | July 1993 |
| | | Site 2 (9-year-old trees) | Eucalyptus grandis | HPM | 29.7 ± 4.53 | 34.3 ± 1.15 | 68.0 | |
| Eamus, O'Grady & Hutley (2000) | North Australian Tropical Transect -Dry site (130°45′E, 12°30′S) -Intermediate site (132°39′E, 14°40′S) - Wet site (133°46′E, 17°07′S) | | Eucalyptus miniate Eucalyptus tetrodonta Eucalyptus latifolia Eucalyptus Capricornia Eucalyptus spp. Eucalyptus terminalis | HPM | 8.8–30.4 4.9–48.7 9.7–48.7 6.1–35.6 6.1–35.6 9.1–41.6 | NA | 16.1 | August–September 1998 (Dry season) |
| | | | | | | | 17.9 | March–April 1999 (Wet season) |
| Dierick & Hölscher (2009) | The Philippines (10°45′55″N, 124°47′25″E) and (10°44′10″N, 124°48′16″E) | | Shorea contorta Parashorea malaanonan Hopea malibato Hopea plagata Swietenia macrophylla. Vitex parviflora Myrica javanica Sandoricum koetjape Durio zibethinus Gmelina arborea | TDP | 18.2 ± 7 12 ± 0.4 11.6 ± 2.4 6.6 ç 1.0 14.6 ± 1.3 20.4 ± 5.5 22.1 ± 3.7 16.3 ± 2.7 19.8 ± 7.3 21.9 ± 4.0 | 16.1 ± 3.5 13.1 ± 1.6 13.3 ± 1.8 9.4 ± 1.2 14.2 ± 1.5 12.7 ± 1.6 11.2 ± 0.6 13.2 ± 1.1 13.8 ± 3.0 18.1 ± 2.4 | 18.4 ± 19.5 10.6 ± 1.2 9.1 ± 8.5 4.0 ± 1.9 25.5 ± 4.1 30.7 ± 14.6 61.7 ± 17.0 32.8 ± 16.5 44.6 ± 18.5 27.6 ± 7.8 | June to August 2006 and July to September 2007 |
| Dierick et al. (2010) | The Philippines (10°45′55″N, 124°47′25″E) and (10°44′10″N, 124°48′16″E) | | Shorea contorta Parashorea malaanonan Hopea malibato Hopea plagata Swietenia macrophylla Vitex parviflora Myrica javanica. Sandoricum koetjape. Durio zibethinus Gmelina arborea | TDP | 18.2 ± 7 12 ± 0.4 11.6 ± 2.4 6.6 ± 1.0 14.6 ± 1.3 20.4 ± 5.5 22.1 ± 3.7 16.3 ± 2.7 19.8 ± 7.3 21.9 ± 4.0 | 16.1 ± 3.5 13.1 ± 1.6 13.3 ± 1.8 9.4 ± 1.2 14.2 ± 1.5 12.7 ± 1.6 11.2 ± 0.6 13.2 ± 1.1 13.8 ± 3.0 18.1 ± 2.4 | 18.4 ± 14.4 10.6 ± 1.1 9.1 ± 6.7 4.0 ± 1.3 25.5 ± 3.6 20.7 ± 9.3 43.2 ± 12.5 23.4 ± 12.6 32.9 ± 14.8 19.8 ± 6.1 | June to August 2006 and July to September 2007 |
| | Indonesia (1.552°S, 120.020°E) | | Theobroma cacao Gliricidia sepium | TDP | 10.1 ± 1.6 15.0 ± 2.5 | 4.5 ± 0.8 10.9 ± 2.1 | 10.0 ± 4.5 13.9 ± 4.1 | February 2007 (Dry season) |
| | Panama (9.317°N, 79.633°W) | | Luehea seemannii Anacardium excelsum Hura crepitans Cedrela odorata Tabebuia rosea | TDP | 11.8 ± 1.6 10.1 ± 0.6 18.0 ± 2.3 12.0 ± 0.6 11.5 ± 1.3 | 8.7 ± 1.0 6.4 ± 0.4 5.4 ± 1.0 11.7 ± 1.1 7.4 ± 0.3 | 13.1 ± 3.6 10.5 ± 2.8 14.6 ± 7.6 9.9 ± 2.2 7.9 ± 0.6 | June to September 2007 (Wet season) |

Ampornpitak et al. (2023), *PeerJ*, DOI 10.7717/peerj.16525

| Reference | Location | | Species | Method | DBH (cm) | Tree height (m) | T (Ld−1) | Period |
|---|---|---|---|---|---|---|---|---|
| *Clulow et al. (2013)* | Nkazana Peat swamp forest site (28°10.176′S, 32°30.070′E) | | *Syzygium cordatum* *Shirakiopsis elliptica* | HPM | NA | 22.5 6.8 | 30–45 2–12 | 4 September 2009 to 4 May 2011 |
| | Dune forest site (28°12.017′S, 32°31.633′E) | | *Drypetes natalensis* *Eugenia natalitia* *Mimusops caffra* | HPM | NA | 4.5 7.5 7.2 | 5–45 2–28 1–4 | |
| *Chen et al. (2014)* | Northwest China (38°11′N, 109°28′E) | | *Ziziphus jujuba* | TDP | 6.69–11.46 | 1.39–1.63 | 12.52–19.47 | May to October 2012 (growth season) |
| | | | | | 0–42.0 7.0–27.0 6.0–17.0 | | | |
| *Cavaleri et al. (2014)* | Lowland wet forest on Hawaii Island (19°42.15′N, 155°2.40′W ) | Invaded forest plots | *Metrosideros polymorpha* *Cecropia obtusifolia* *Macaranga mappa* *Melastoma septemnervium* | TDP | 9.7.0–19.0 | NA | 2–25 | February to November 2008 |
| | | Removal plots | *Metrosideros polymorpha* | TDP | 9.0–42.0 | NA | 5–43 | |
| *Hardanto et al. (2017)* | Rubber monoculture | | **Rubber trees** *Hevea brasiliensis* | TDP | 20.3 ± 0.6 | 13.4 ± 0.4 | 25.6 ± 3.7 | June to August 2013 |
| | Jungle rubber | | **Rubber trees** *Hevea brasiliensis* | TDP | 17.8 ± 0.5 | 140. ± 0.5 | 24.1 ± 4.2 | |
| | | | **Admixed native trees** *Cratoxylum sumatranum Callerya atropurpurea Ixonanthes petiolaris Santiria griffithii Macaranga cf. sumatrana Artocarpus nitidus Alstonia angustifolia Streblus elongates Artocarpus integer Porterandia anisophylla Timonius wallichianus* | TDP | 18.03 ± 0.3 | 14.0 ± 0.2 | 26.7 ± 2.2 | |

**Table 3** (*continued*)

| Reference | Location | Species | Method | DBH (cm) | Tree height (m) | $T$ (Ld−1) | Period |
|---|---|---|---|---|---|---|---|
| *Brum et al. (2018)* | Mature lowland Amazon forest (2°31′0S, 48°53′W) | Canopy trees | HPM | 30–109 | NA | 68 ± 87 | October 2015 to April 2016 |
| | | Subcanopy trees | HPM | 10–30 | NA | 11 ± 10.04 | |

**Notes.**

Mean values (±SD if available) are presented. The abbreviations are DBH, diameter at breast height (−1.3 m above ground); $T$, tree water use (Ld−1); TDP, thermal dissipation probes; HPM, heat pulse method. NA indicates not available data.

The response pattern of saturating exponential function of $T$ with VPD found in this study is similar to the one observed in various tree species in a wide range of tropical forests, including a lowland tropical forest of Central and northern South America (*Meinzer et al., 1993*), a primary lowland tropical forest in eastern Amazon (*Brum et al., 2018*) and a per-humid tropical forest of Central Sulawesi in Indonesia (*Horna et al., 2011*). A previous study showed that tree transpiration strongly increases with rising VPD under high soil water availability; however, such response may become weaker or disappear when soil moisture is lower, depending on tree species (*Butz et al., 2018*). In general, we observed similar responses of $T$ to VPD under wet and dry conditions with stronger responses in the former; whereas no responses were detected when trees experienced moderate soil moisture. Under dry conditions, our results indicate that $Sa$ was sensitive to increasing VPD while $Ai$ can maintain their water use rate regardless of changes in VPD, regardless of tree size. This implies that $Ai$ may be more tolerant to drought than $Sa$ and may have strong control over their water use under low soil moisture, regardless of tree size, which can prevent it from negative effects from droughts. This result agreed with a previous study that investigated the drought tolerance of both species in this site (*Unawong et al., 2022*). Based on tree hydraulic measurement, the study reported that xylem pressure at 50% loss of hydraulic conductivity ($P_{50}$) of $Ai$ and $Sa$ were $-5.97$ and $-4.71$ MPa, respectively. It is implied that species with lower $P_{50}$ have greater resistance to embolisms, thus allowing better adaptation to environments where water stress frequently occurs (*Maherali, Pockman & Jackson, 2004*). When comparing $T$ in different size classes of $Sa$, large trees were less sensitive to rising VPD at lower VPD ranges. The less sensitivity of large trees to rising VPD leads to a slower decrease in water consumption rate to save water than small trees, resulting in potentially greater vulnerability to hydraulic failure during drought in large trees. Previous studies have shown size-dependent sensitivity to droughts in many ecosystems. A synthetic study using data on tree growth and mortality, which were collected during 40 drought events in forests worldwide, showed that droughts consistently exerted negative impacts on the growth and mortality rates of larger trees (*Bennett et al., 2015*). Greater vulnerability of large trees to drought could be affected by the higher exposure to radiation and atmospheric demand because of increasing tree height (*Roberts, Cabral & Aguiar, 1990*; *Nepstad et al., 2007*). Moreover, large trees have to transport water to greater heights, which is against the effects of gravity, thus facing greater hydraulic failure (*Ryan, Phillips & Bond, 2006*; *Zhang et al., 2009*). Thus, large $Sa$ may be at higher risk of hydraulic failure when drought is more pronounced, plausibly leading to increasing mortality rates (*Choat et al., 2018*). At moderate soil water, the results indicated that both species could maintain their tree water use, regardless of tree size. Under high soil moisture conditions, the $T$ of both species in both sizes also followed the saturating exponential pattern as in the case of low soil moisture conditions. However, the sensitivity of increasing $T$ at low VPD was different between sizes. In both species, $T$ of large sizes was less sensitive to rising VPD than small ones. In other words, when the air becomes dry, small trees may decrease water consumption rate faster to save water than large trees. This may be partly because small trees mainly use shallow soil water whereas large trees can access water from deeper soil (*Brum et al., 2018*), allowing less sensitivity to droughts in large trees. Nevertheless,
further studies that investigate water-source partitioning of different tree species in the same forest (*e.g.*, *Hasselquist, Allen & Santiago, 2010*; *Wang et al., 2020*), tracing isotopic signals of water from various soil layers to the stems, may be performed to confirm these results.

### Implications for environmental management

The results from this study imply that *Sa* may provide ecosystem disservice in dry areas due to its high water consumption which results in low water supply for the downstream community, but it may slow down runoff in the region that experiences heavy precipitation. In contrast, *Ai* may provide ecosystem benefits by conservatively using water, even under drought conditions, but may increase runoff when storms come with high rainfall. Another implication is that *Ai* may be suitable for reforestation in the area where droughts frequently occur in downstream ecosystems through its conservative water-use behavior, thus maintaining runoff from the forests during drought. Moreover, because *Ai* showed relatively constant water use regardless of tree size, the species would still provide such benefits to the ecosystems even when it grows larger in the future. In contrast, *Sa* may be appropriate for reforestation in the area with frequent floods because it has high water consumption during high water availability which may decelerate runoff from forests into downstream ecosystems. This would benefit the downstream ecosystems when storms occur. Regardless, mixed planting species seem to be suitable for reforestation in the areas where extreme events do not frequently occur because both species can maintain their water use at moderate soil moisture regardless of tree size, therefore preventing the depletion of soil water availability. In addition, mixed planting species could reduce the competition for limited water resources because the differences in root structures of different tree species lead to less competition for water (*Schwendenmann et al., 2015*). Nevertheless, reforestation projects should emphasize the use of native species to avoid competition with other native trees on the site (*Hooper, Condit & Legendre, 2002*).

## CONCLUSIONS

We estimated tree water use ($T$) of dominant tree species including *Syzygium antisepticum* (*Sa*) and *Adinandra integerrima* (*Ai*) in a secondary tropical forest in Khao Yai National Park from sap flux density ($J_s$) which was continuously monitored with custom-made thermal dissipation probes and compared $T$ of both species in different tree size classes. In addition, we evaluated the responses of $T$ to environmental factors of both species in different tree size classes. The results showed that $T$ of *Sa* was significantly higher than *Ai* and that large trees had higher $T$ than small ones which was related to relatively lower sapwood area in the small trees. Further analysis of the response patterns of $T$ showed that *Sa* was more sensitive to increasing VPD than *Ai* while *Ai* can maintain their water use regardless of tree size under low soil moisture. This implies that *Ai* may be able to cope with the negative effects of droughts and retain such capacity when they grow. With ample soil moisture, both species can maintain their tree water use regardless of tree size. When soil moisture becomes high, the $T$ of both species in both sizes increases with rising VPD and then saturated at high VPD. Nevertheless, $T$ of both species in large size

was less sensitive to rising VPD than in small size which may be explained by the deeper access to groundwater in large trees. For the implications for management, our results suggest that *Ai* may be suitable for reforestation in the area where droughts frequently occur in the downstream ecosystem through its conservative water-use behavior and may benefit downstream ecosystems with continuous runoff from the forest despite droughts. Moreover, *Ai* has conservative water-use behavior regardless of tree size. Thus, *Ai* would still provide these benefits to ecosystems when they grow larger in the future. In contrast, *Sa* seems suitable for reforestation in the area with frequent floods because it has high water consumption during high water availability which may slow down runoff from forest into downstream ecosystems when storms come. However, mixed planting species may be suitable for reforestation in areas where extreme events do not frequently occur because both species can maintain their water use at moderate soil moisture regardless of tree size which prevents the depletion of soil water availability. In this case, depending on the purposes of reforestation, *Sa* and *Ai* may provide either benefits or negative effects to the ecosystems. In conclusion, this study highlights the dependency of responses of *T* to environmental conditions on tree species and size. Such information would benefit the selection of tree species for reforestation that could adapt well to certain environments and support policy design on the management of tropical forests and water resources. Nevertheless, a further study involving additional field measurements of the physiological parameters of trees, such as root depth, is needed to support the proposed findings.

## ACKNOWLEDGEMENTS

We would like to thank Miss Nichaphan Kasikam, Mr. Rathasart Somnuk, Miss Jutawan Moonongsang, and Miss Jeerapat Thawjaturat for field assistance.

### Funding

This research was funded by the Thailand Science Research and Innovation Fund Chulalongkorn University (DIS66230005). Ratchanon Ampornpitak was supported by Thailand Graduate Institute of Science and Technology (TGIST), National Science and Technology Development Agency through a postgraduate scholarship (SCA-CO-2564-14439-TH) and the International Postgraduate Program in Hazardous Substance and Environmental Management, Graduate School, Chulalongkorn University. The funders had no role in study design, data collection and analysis, decision to publish, or preparation of the manuscript.

### Grant Disclosures

The following grant information was disclosed by the authors:
Thailand Science Research and Innovation Fund Chulalongkorn University: DIS66230005.
Ratchanon Ampornpitak was supported by Thailand Graduate Institute of Science and Technology (TGIST), National Science and Technology Development Agency through a postgraduate scholarship: SCA-CO-2564-14439-TH.

International Postgraduate Program in Hazardous Substance and Environmental Management, Graduate School, Chulalongkorn University.

## Competing Interests
The authors declare there are no competing interests.

## Author Contributions
- Ratchanon Ampornpitak performed the experiments, analyzed the data, prepared figures and/or tables, authored or reviewed drafts of the article, and approved the final draft.
- Anuttara Nathalang analyzed the data, authored or reviewed drafts of the article, provide facility at the field site, and approved the final draft.
- Pantana Tor-ngern conceived and designed the experiments, performed the experiments, analyzed the data, authored or reviewed drafts of the article, and approved the final draft.

## Data Availability
The raw data are available in the Supplementary File.

## Supplemental Information
Supplemental information for this article can be found online at http://dx.doi.org/10.7717/peerj.16525#supplemental-information.

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
