# Peer review of "Water-use characteristics of Syzygium antisepticum and Adinandra integerrima in a secondary forest of Khao Yai National Park in Thailand with implications for environmental management"

_PeerJ, doi:10.7717/peerj.16525_

## Round 0.1 · original submission · Major Revisions

I hope this message finds you well. I would like to express my appreciation for your submission. Your work contains valuable information for both the academic community and reforestation practitioners.

However, after careful evaluation, it is my assessment that the current version of the manuscript does not meet the criteria for publication. Several areas require further elaboration and correction to ensure the quality and rigor expected for publication.

Specifically, given the pivotal role of soil moisture in your study, I recommend providing additional information about soil properties, particularly texture and soil depth. This will be crucial in effectively interpreting the soil moisture data.

Additionally, please clarify how the average soil moisture was computed for determining REW. Moreover, could you please provide details on the average soil depth and estimated rooting depth of these two tree species? Please discuss the level of confidence you have in the meaningfulness of your soil moisture data from the upper 30 cm with respect to its relevance to transpiration response.

Additionally, please address the sample size and provide details regarding the statistical analysis when the data is further divided into size groups.

Regarding the comparison of water use among different species, it is worth noting that the individual trees of one species are significantly larger than those of the other species. While it is not surprising that the larger tree would use more water on a per-tree basis, it is also important to consider that the tree density is typically lower for larger trees. As a result, a forest with small trees can use as much water if the tree density is high enough. In this context, I suggest comparing ecosystem evapotranspiration (ET) (in mm per unit area) to account for variations in tree density. In cases with a substantial difference in DBH, you may find that sapflow density provides a more meaningful metric for specific water usage (see Torquato et al., 2020 for reference).

While the language is generally comprehensible, there are areas that require correction and improvement. I have marked some of these places for your reference during a quick read-through.

I kindly request that you address these comments and submit a revised version of the manuscript for further consideration. It is important for you to know that the resubmission can be rejected or may need to undergo another round of review depending on how well all those issues are addressed in the resubmission.

Thank you for your time and effort in addressing these concerns. We look forward to receiving your revised submission.

Reviewer 1 ·

Basic reporting

No comment

Experimental design

I think the methods to find out tree water use (T) and vapor pressure deficit (VPD) should be elaborated in a reproducible way.

Validity of the findings

Why authors selected S. antisepticum and A. integirrima for this study? In fact, both have different sizes (heights). Please justify this point with a reference.

Additional comments

This is a nice study covering a vital aspect of agroforestry. Additionally, it is well organized and concluded with a clear message/result. Based on that, both tree plants might be suggested for different ecosystems (drought/marginal/flooded). I have few suggestions or questions. If authors think they are relative, then please address them.
Some authors have used different characteristics to check water use potential / efficiency of tree plants. For example, hydrogen (δ2H) and oxygen (δ18O) isotopes for xylem and soil water uptake. Moreover, δ13C values for plant leaves (Reference: https://www.sciencedirect.com/science/article/abs/pii/S0168192320301222). Have you considered that for your study? Or you did not need to go through that. Please justify these points.

Reviewer 2 ·

Basic reporting

The manuscript clear unambiguous and professional English. Introduction part is well organized, show the novelty of their work, and demonstrate how work fits into broader field of knowledge. Literature reviews are appropriate referenced and sufficient context provided. Figures, and tables are relevant to the context.
Self-contained with relevant results to hypotheses. The manuscript clearly outlined the objectives and aim of the study and included knowledge on the subject. Well done!

Experimental design

The author clearly explains with sufficient information for their experiment design. However, I have some comment on this section for them to clarify. Research question also well defined.

Validity of the findings

Impact and novelty with meaningful replication encouraged. All underlying data have been provided, statistically sound and controlled. Conclusions are well stated with linked to the original research question.

Additional comments

They did a good job. This finding can be implemented to the forestry industry as well.

Annotated reviews are not available for download in order to protect the identity of reviewers who chose to remain anonymous.

---

## Round 0.2 · Minor Revisions

Please see my suggested changes, especially on the abstract, but also other comments (pdf attached). I also pointed out some grammatical issues in the manuscript body for your attention.

Reviewer 1 ·

Basic reporting

no comment

Experimental design

no comment

Validity of the findings

no comment

Additional comments

You have addressed my all queries. Good luck

---

## Round 0.3 · accepted · Accept

Dear Dr. Tor-ngern,

I have reviewed your revision and I am pleased with the current version. Your work contributes water use information for two additional tree species to the growing list of tropical tree species. I believe this will be very useful for the ecohydrological community. Thank you and your team for your efforts.

Chris Zou
Professor in Ecohydrology
Oklahoma State University
Academic Editor
PeerJ Life & Environment